# Biology and Function of miR159 in Plants

**DOI:** 10.3390/plants8080255

**Published:** 2019-07-30

**Authors:** Anthony A. Millar, Allan Lohe, Gigi Wong

**Affiliations:** Division of Plant Science, Research School of Biology, The Australian National University, Canberra ACT 2601, Australia

**Keywords:** miR159, *GAMYB*, programmed cell death, aleurone, tapetum, vegetative growth, flowering

## Abstract

MicroR159 (miR159) is ancient, being present in the majority of land plants where it targets a class of regulatory genes called *GAMYB* or *GAMYB-like* via highly conserved miR159-binding sites. These *GAMYB* genes encode R2R3 MYB domain transcription factors that transduce the gibberellin (GA) signal in the seed aleurone and the anther tapetum. Here, *GAMYB* plays a conserved role in promoting the programmed cell death of these tissues, where miR159 function appears weak. By contrast, *GAMYB* is not involved in GA-signaling in vegetative tissues, but rather its expression is deleterious, leading to the inhibition of growth and development. Here, the major function of miR159 is to mediate strong silencing of *GAMYB* to enable normal growth. Highlighting this requirement of strong silencing are conserved RNA secondary structures associated with the miR159-binding site in *GAMYB* mRNA that promotes miR159-mediated repression. Although the miR159-*GAMYB* pathway in vegetative tissues has been implicated in a number of different functions, presently no conserved role for this pathway has emerged. We will review the current knowledge of the different proposed functions of miR159, and how this ancient pathway has been used as a model to help form our understanding of miRNA biology in plants.

## 1. Introduction

Associated with the emergence and diversification of land plants, is a core set of conserved microRNA (miRNA) families that arose early in terrestrial plant evolution and which are conserved in modern day plant species [1]. This conservation implies these endogenous gene regulators are fundamental to plant biology and have been indispensable for the conquest of plant life on land. One such core family is microR159 (miR159), which has now been extensively studied in multiple, diverse plant species. In this review, we will highlight the major functions identified for miR159, its use as a model for gaining greater insights into miRNA biology in general, and finally highlight the many outstanding questions surrounding this ancient gene regulator.

## 2. MiR159 is Strongly Conserved and Highly Abundant Throughout the Plant Kingdom

Surveys of the many deep sequencing experiments on the small RNA fractions of plants, find miR159 ubiquitously present as a 21 nucleotide (nt) miRNA in all eudicots and monocotyledonous plants examined [2], and present in the majority of basal angiosperms, gymnosperms, ferns, and lycopods examined [3]. It has either been classified as a Class I (ubiquitous) or a Class II (present in most taxonomic groups) miRNA [2,3]. There is some uncertainty regarding whether miR159 is present in Bryophytes, where it is generally regarded as being absent [4], but it has been reported in a liverwort [5]. However, the reported miR159 sequence was only 18 nt long, suggesting it may not be a genuine miR159 homologue, so further analyses will be required to resolve this. Nevertheless, it is apparent that miR159 arose early in basal land plants and has been strongly conserved henceforth.

MiR159 fits the curious observation that the stronger the miRNA is conserved, the greater its expression or abundance [2]. This was derived from a multitude of small RNA-sequencing experiments from a wide diversity of plant species, where miR159 is often among the most abundant small RNA species (e.g., [6,7,8,9,10]). Additionally, highly similar miR159 isoforms are present in most land plants (Figure 1; [11]), so the sequence of this canonical miR159 appears to have remained fixed for 100s of millions of years. Nevertheless, like most miRNA families, considerable variation exists within small RNAs defined as miR159, with most plant species containing multiple family members that encode identical or highly similar isoforms, or “isomiRs”, that differ by one to several nucleotides. For example, maize has 11 different *MIR159* loci, encoding four different miR159 isoforms [12]. For the most part, nucleotide variation occurs at the extremities of the miRNA, at positions considered less important for its specificity [13]. This is the case for the three different miR159 isoforms found in Arabidopsis that vary by 1–2 nucleotides; however, as these isoforms appear functionally redundant, this variation unlikely impacts which genes they target for repression [14,15]. Some species have even more variant miR159 isoforms (e.g., poplar; grape, soybean, and maize with 3–5 sequence variations [16]), so whether these miR159 variants have sub-functionalized to regulate different targets remains a possibility. Indeed, the ancient miRNA miR319 is closely related to miR159. In Arabidopsis, these two families are identical at 17 of 21 nucleotide positions, but have distinct target genes, demonstrating their sub-functionalization [17]. Their similarity extends to their primary-*MIRNA* precursors, where *pri-MIR159* and *pri-MIR319* are both unusually long fold-back structures that are processed in a non-canonical loop-to-base direction [18]. Phylogenetic analysis of primary-*MIRNA* precursor sequences of these families supports a common origin of miR319 and miR159, with the likelihood that miR159 has arisen and specialized from miR319 in basal land plants [19].

## 3. GAMYB and GAMYB-like Genes are the Only Conserved Targets of miR159

Core to understanding the function of a miRNA is the identification of the genes that they target. A clear and recurrent theme is that miR159 targets a family of genes encoding R2R3 MYB transcription factors referred to as “GAMYB” or “GAMYB-like”. Similar to the conservation of miR159, *GAMYB*-homologues with a highly conserved miR159 binding site are found in most lineages of land plants (Figure 1). This extends to basal plants such as lycopods (e.g., *Selaginella moellendorffii*), moss (e.g., *Physcomitrella patens*), and the liverwort *Marchantia polymorpha* [20,21]. However, in *Marchantia* it appears that the *GAMYB* homologue is not regulated by miR159, but rather by miR319 [21,22]. Even in Arabidopsis, miR319 can regulate the *GAMYB* targets [17]. However, as miR319 in Arabidopsis is narrowly and weakly expressed compared to the widely and abundantly expressed miR159, miR319-mediated regulation of the *GAMYB-like* genes is insignificant relative to miR159-mediated regulation [17]. This makes miR159 functionally specific for the *GAMYB-like* targets, whereas miR319 is functionally specific for genes encoding another class of transcription factors, the *TCP* family, which miR159 is unable to regulate [17,23]. Therefore, it appears likely the more specific miR159 has arisen from miR319 in basal land plants, sub-functionalizing to become specific for the *GAMYB-like* genes. Although there is sequence variation in both the miR159 and its binding site within the *GAMYB-like* genes, both have appeared to have become fixed, arguing that this ancient miR159-*GAMYB* target relationship is critical for the life of land plants [24].

Strong experimental evidence supports the prediction of conserved miR159-mediated regulation of *GAMYB*. Firstly, degradome analysis from multiple diverse species confidently identifies *GAMYB* homologues are being regulated via a miR159-mediated cleavage mechanism. Although this analysis only detects targets regulated by the miRNA-guided cleavage mechanism (and not the translational-repression mechanism), functionally important targets appear to be preferentially detected [25]. Degradome experiments have been mainly performed on higher plants, including eudicots such as Arabidopsis [25], soybean [26], cotton [27], tomato [28], orchids [29], and peach [30]; also monocots such as wheat [31], rice [32], and barley [33], among many others, all of which experimentally validate *GAMYB* homologues as targets of miR159. Although many of these degradome experiments also pick up other target genes (e.g., [27,31]), these other targets are diverse in their identity and do not appear to be broadly conserved miR159 targets; i.e., they are not identified in degradomes from multiple diverse species. This argues that although miR159 may regulate additional targets, this does not appear to be at the expense of its main target, *GAMYB*. For instance, in tomato, miR159 has acquired a novel target, a gene that encodes a protein with a NOZZLE-like domain, and this miR159-mediated regulation is important for tomato development [34]. However, miR159 still regulates *GAMYB-like* genes in tomato, which is important for fruit development [35].

## 4. The miR159-*GAMYB* Pathway in Arabidopsis

The Arabidopsis miR159 family has been extensively studied as a model for plant miRNA-mediated gene regulation (Figure 2). Arabidopsis has three *MIR159* genes (*MIR159a*, *MIR159b*, and *MIR159c*), each encoding a distinct isoform that differ from one another by 1–2 nucleotides [8]. Examination of their expression domains with promoter: GUS constructs found *MIR159a: GUS* and *MIR159b: GUS* had highly similar expression patterns, being broadly expressed throughout the plant, but strongest in shoot and root meristematic regions [14]. By contrast, the expression domain of a *MIR159c: GUS* reporter gene was much narrower, being restricted mainly to anthers and the shoot apical region [15], suggesting sub-functionalization. Regarding their level of expression, both deep sequencing and qPCR has found miR159a to be the most abundant family member, with miR159c being very weakly expressed [8,15]. To investigate their function, T-DNA loss-of-function mutant alleles were generated for each gene, however, none of the single *mir159* mutant plants displayed any phenotypic defects [14,15]. However, consistent with the highly similar expression domains, miR159a and miR159b were demonstrated to be functionally redundant, as a double *mir159a.mir159b* (*mir159ab*) mutant displayed severe growth and developmental defects, most notably a smaller rosette with upwardly curled leaves (Figure 2; [14]). As a triple *mir159abc* mutant appeared indistinguishable from *mir159ab*, this and other data suggested miR159c in Arabidopsis has little to no activity and possibly corresponds to a pseudogene [15,17]. This is one of the few instances in Arabidopsis where T-DNA mutants have been identified and combined for all members of a miRNA family, and the *mir159ab* and *mir159abc* mutants have been used extensively in the functional characterization of miR159.

A bioinformatic search of miR159 targets in Arabidopsis using the standard target prediction program psRNATarget, identifies almost 100 potential miR159 targets with four or less mismatches [40]. The top twenty targets are shown in Table 1, which includes eight *MYB* genes with highly conserved miR159 binding sites [15]. By contrast, the non-*MYB* genes are highly diverse and their miR159-binding sites do not appear conserved [15]. Of the conserved *MYB* targets, seven are *GAMYB-like* genes (*MYB33*, *MYB65*, *MYB81*, *MYB97*, *MYB101*, *MYB104,* and *MYB120*), and the other is non-*GAMYB-like* gene, *DUO1* (*DUO POLLEN1*), which has a conserved miR159 binding site at a position distinct from the *GAMYB-like* genes [17]. Despite the fact that miR159-mediated cleavage products can be isolated for many of these predicted targets (Table 1), transcript profiling of the *mir159ab* mutant only identified two genes that appeared strongly de-regulated, the *GAMYB-like* targets, *MYB33* and *MYB65* (Table 1). This de-regulation resulted in *MYB33* and *MYB65* being strongly expressed throughout the plant [14,41]. Consistently, the only genes detected in multiple degradome analyses were *MYB33* and *MYB65* (Table 1) [25,42]. Eliminating the expression of these genes via the introduction of *myb33* and *myb65* loss-of-function alleles, suppressed all vegetative phenotypic defects of *mir159ab*, as a *mir159ab.myb33*.*myb65* quadruple mutant appeared indistinguishable from wild-type, other than male sterility [14]. The phenotype of male sterility is the only apparent defect of *myb33.myb65* plants, as *MYB33* and *MYB65* are two redundant genes that facilitate anther development (Figure 2) [43].

These genetic experiments demonstrated the major role of miR159 in Arabidopsis as being the widespread suppression of *MYB33* and *MYB65* expression, whose activity has severe deleterious impacts on plant growth and development, including stunted growth and curled leaves (Figure 2). The experiments also defined the functional specificity of miR159 in Arabidopsis as being *MYB33* and *MYB65* [14]. Supporting this is the expression of either a miR159-resistant *MYB33* or miR159-resistant *MYB65* transgene, both of which can phenocopy the *mir159ab* mutant [14,23,44]. This much narrower functional specificity compared to the bioinformatic prediction of many more targets is a common theme in miRNA biology, where both in animal and plants, pleiotropic defects of miRNA mutants can be suppressed via the repression of one-two target genes, despite bioinformatic programs predicting many targets with conserved miRNA binding sites [24,45]. Partially explaining this phenomenon for miR159, many of the bioinformatically predicted miR159 targets appear to have transcriptional domains that are mutually exclusive to that of miR159. Hence, the miRNA and targets are physically separated spatially and/or temporally preventing interaction (Figure 2) [14,15].

Despite their deleterious impact on vegetative growth, *MYB33* and *MYB65* appear ubiquitously transcribed throughout the plant, but only to be strongly and ubiquitously silenced, other than in seeds and anthers (Figure 2) [41,46]. There are multiple lines of evidence supporting this claim; (1) the vegetative phenotype of *myb33.myb65* appears indistinguishable from wild-type; (2) the transcriptome profiles of shoot apical regions of wild-type versus *myb33.myb65* appear indistinguishable; (3) the expression of a *MYB33: GUS* transgene is undetectable in GUS-stained vegetative tissues, but a miR159 resistant version of the reporter gene, *mMYB33: GUS,* is widely and strongly expressed (Figure 2) [41]. The efficiency of this silencing is highlighted by the *mir159a* single mutant; although deep sequencing demonstrates miR159a is the predominant isoform (e.g., miR159a–6621 reads, miR159b–982 reads [8]), the *mir159a* mutant appears indistinguishable from wild-type [14], implying strong reductions in miR159 levels do not impact the silencing of *MYB33*/*MYB65*. Conversely, overexpression of a wild-type *MYB33* gene fails to result in any phenotypic defects [47]. Although these *MYB33* overexpressing Arabidopsis plants have high *MYB33* mRNA levels, they do not exhibit any phenotypic defects, indicating miR159 also represses expression of *MYB33*/*MYB65* mRNA via a translational repression mechanism [47]. The importance of this mechanism was shown via the complementation of *mir159ab* with a mutated miR159 variant that had two mismatches to *MYB33*/*MYB65* at the cleavage site; although this attenuated cleavage, this miR159 variant could still potently silence *MYB33*/*MYB65* [47]. Therefore, these combined silencing mechanisms ensure *MYB33*/*MYB65* are strongly repressed in vegetative tissues.

## 5. Conserved RNA Secondary Structures in *MYB33*/*65* Promote miR159-Mediated Silencing

Highlighting this efficient silencing were miR159 efficacy assays performed on the various Arabidopsis *MYB* targets [44]. Here, it was demonstrated that *MYB33* and *MYB65* were very sensitive targets of miR159, being strongly silenced. In contrast, the other *MYB* genes (*MYB81*, *MYB97*, *MYB101*, *MYB104*, and *DUO1*), were poorly silenced by miR159. As all these *MYB* targets had highly complementary miR159 binding sites, it implies factors other than complementarity must be contributing to this differential miR159-mediated silencing [44]. Correlating with this difference, is a predicted RNA secondary structure that abuts the miR159 binding site of *MYB33* and *MYB65*, but which is absent in the poorly regulated targets (Figure 3; also see [44] for RNA secondary structures of the various Arabidopsis *GAMYB-like* genes). To determine the significance of this in silico predicted RNA structure, a structure/function analysis was performed. Mutation of this structure within the *MYB33* context attenuates silencing, whereas the restoration of the structure, although with a different primary nucleotide sequence, restores strong silencing of *MYB33* [44]. Therefore, this demonstrates that this RNA secondary structure facilitates *MYB33* and *MYB65* silencing, earmarking them as functional targets of miR159. It argues that a fully functional miR159 target site of *MYB33*/*MYB65* encompasses nucleotides beyond that of the binding site.

Further evidence of the importance of this RNA secondary structure is its strong conservation in *GAMYB-like* homologues throughout the plant kingdom (Figure 3), as the nucleotides that correspond to the stems of the RNA secondary structures are conserved in *GAMYB* homologues of eudicots, monocots, and basal angiosperms, such as Amborella [44]. This indicates this structure is part of the miR159-*GAMYB* regulatory relationship and that the mechanism of regulation is likely more complex than miRNA-binding site complementarity alone. Given that so many miRNA-target relationships are ancient, it will be interesting to investigate how many other miRNA targets have conserved RNA elements associated with their miRNA binding sites, as these ancient regulatory relationships have had 100s of millions of years to evolve greater regulatory complexity.

In Arabidopsis, not only do the poorly regulated *GAMYB-like* genes lack this conserved RNA structure, but they have highly specific transcriptional domains, predominantly in seeds and anthers, where miR159 activity appears attenuated or absent [14,15,49,50]. Therefore, strong selection of this RNA secondary structural element may have occurred for *GAMYB* homologues that are transcribed in vegetative tissues and require strong miR159-mediated silencing to prevent deleterious outcomes [44]. Investigating whether this also applies to other species with multiple *GAMYB* homologues will be interesting to follow up. Finally, the efficacy of miR159-mediated silencing of *MYB33* in Arabidopsis varies between tissues, being strong in the rosette, but weak in the seed [48]. As RNA secondary structures are dynamic in vivo, they may be operating as a riboswitch, with certain formations facilitating silencing, and others attenuating silencing. It will be interesting to determine whether the conformation of the RNA secondary structure changes between tissues, controlling the ability of miR159 to silence *MYB33*.

## 6. The Function of miR159-*MYB* Pathway in Plant Development

The functional role of the miR159-*GAMYB* pathway has been studied in numerous plant species, and this is summarized in Table 2.

### 6.1. A Role in Male Reproductive Development

The GAMYB/GAMYB-like family of transcription factors is found throughout the plant kingdom, where they share high sequence similarity in their R2R3 DNA-binding domains located towards the N-terminal region, but are much more diverse in their C-terminal regions [20]. Nevertheless, the functions of these *GAMYB* homologues appear to be highly conserved, as *GAMYB* homologues from *Lycopods* or Bryophytes can partially complement a *gamyb-2* rice mutant [20], or a cucumber *GAMYB* can complement the male sterile phenotype of the Arabidopsis *myb33.myb65* mutant [68]. Hence, despite the sequence diversity of the C-terminal regions, this complementation of distant species argues the biochemical function of GAMYB has been strongly conserved.

To date, a role in male reproductive development appears the clearest function for *GAMYB* [70]. Inhibition of its activity perturbs male development, whether in basal plants such as lycopods (*Selaginella moellendorffii*) or bryophytes (moss-*Physcomitrella patens*) [20], or in higher flowering plants, such as rice [64] or Arabidopsis [43]. Moreover, GAMYB was shown to positively regulate the *CYP703* gene, which is required for male development in both basal and higher plants [20,71]. It appears this *GAMYB*-*CYP703* pathway arose very early in land plant development, and then has come under the control of gibberellin (GA) in lycopods, likely explaining why male reproductive development in plants is under the control of GA [20,71]. It was speculated that the GA regulation of the *GAMYB*-*CYP703* pathway was a step in the evolution of the sporophyte-dominated life cycle, which requires greater regulatory control for its more complex reproductive system [20].

Consistently, there have been many reports of plants with multiple *GAMYB* homologues for which at least one is strongly transcribed in anthers (e.g., *CsGAMYB1* in cucumber, [68]; *TaGAMYB1* in wheat, [66]; *HvGAMYB* in barley, [65]), or is anther-specific (e.g., *PtrMYB012* in poplar, [16]; *MYB97*, *MYB120* in Arabidopsis, [49]), many of which are positively regulated by GA. Inhibition of GAMYB activity perturbs programmed cell death (PCD) in the anther tapetum [71], where in both a rice *gamyb* mutant, and the Arabidopsis *myb33*.*myb65* mutant, the tapetum fails to degenerate, resulting in hypertrophy, leading to male sterility [43,64,71]. Additionally, *MYB33* and *MYB65* in Arabidopsis are also required for the formation of the radial microtubule array surrounding nuclei immediately following meiosis II [72]. In the *myb33.myb65* mutant, the resulting defects in male meiotic cytokinesis produce diploid pollen with a defective pollen wall morphology [72]. However, the role that miR159 plays in regulating GAMYB expression in male development is unclear, where it may be fine-tuning expression or preventing expression occurring in particular cell layers, but for most species, this is yet to be resolved. In rice anthers, miR159 and *GAMYB* are co-expressed, suggesting potential fine-tuning of *GAMYB* expression [63].

In Arabidopsis, the role of the miR159-*GAMYB* pathway and its interaction with the miR319-*TCP* pathway in flower maturation has been investigated using miRNA loss-of-function *MIM159* and *MIM319* transgenic plants, both of which display multiple pleiotropic defects [59]. In sepals, petals and anthers of these plants, it was found that the GAMYB and TCP proteins are expressed and directly interact to regulate another miRNA, miR167, which creates a miR159-miR319-miR167 network. It is proposed that the function of miR159/miR319 is to dampen MYB/TCP expression, resulting in low miR167, and hence enabling strong ARF6/8 expression, which in turn regulates many genes required for floral development including that of auxin signaling [59]. However, in wild-type plants, it appears *MYB33*/*65* expression in flowers is restricted to anthers, and the *myb33.myb65* mutant only displays anther defects [43]. Therefore, in wild-type, it appears the role of miR159 in flowers is to strongly repress the *MYB* genes in sepals and petals to prevent strong expression of miR167 to ultimately enable strong ARF6/8 expression.

MiR159 is present in pollen where it has a crucial role in fertility [51]. It has been known for some time that sperm cell entry alone triggers central cell division, suggesting that the male genome and/or unknown factors transmitted by the sperm control the initiation of endosperm development. Unexpectedly, the central cell usually fails to initiate division after pollination by *mir159abc* mutants, or stops dividing after one or two divisions, resulting in reduced seed set. It was found that both *MYB33* and *MYB65* are highly expressed in the central cell of the embryo sac before fertilization, but after fertilization, both transcripts are rapidly cleared from the central cell and the endosperm initiates development. It was observed that *MYB33* and *MYB65* transcripts are not cleared in pollinations with *mir159abc* pollen, suggesting that miR159 in pollen is transmitted to the central cell by fertilization where it degrades *MYB33* and *MYB65* transcripts [51]. Thus, miR159 has a paternal effect on seed development: miR159 carried in pollen abolishes central cell repression after fertilization permitting endosperm nuclear divisions [51]. Loss of maternal miR159 also results in seed defects but these defects are less severe on seed set than loss of paternal miR159, and the mechanism of this maternal effect is unknown.

### 6.2. A Role in Seed Development

GAMYB was first identified as a GA signaling component in the barley aleurone, hence giving these MYB genes their name “GA”MYB [73]. Here, GAMYB positively transduces the GA signal to activate expression of α-amylase and other hydrolytic enzymes [74], as well as promote PCD in the aleurone [75]. This latter function appears conserved in Arabidopsis, as a *myb33.myb65.myb101* triple mutant has attenuated vacuolation in aleurone cells, a PCD-mediated process that is positively regulated by GA [41]. Therefore, a conserved role in PCD in the aleurone and tapetum in both monocots and dicots is currently a unifying function for these GAMYB transcription factors. Curiously, both these tissues are single cell layers that provide nutrients upon death to the embryo (aleurone) or pollen (tapetum). It is possible other *GAMYB-like* genes may play similar roles in terms of inhibiting growth and promoting cell death. For example, *MYB97*, *MYB101*, and *MYB120* are all expressed in the pollen tube, and in a *myb97.myb101.myb120* mutant, the pollen tube fails to undergo growth arrest and then fails to degenerate in order to release the sperm cells to the ovules [49,50].

Many downstream targets of the miR159-*GAMYB* pathway in Arabidopsis support a role in PCD. Micro-array analysis on the shoot apical region of *mir159ab* plants found that of the 166 up-regulated genes, many appeared aleurone related [41]. Many of these aleurone related genes were also down-regulated in *myb33.myb65.myb101* seeds, making them strong candidates of being downstream of *GAMYB* activity [41]. This includes the most up-regulated gene in *mir159ab*, *CYSTEINE PROTEINASE 1* (*CP1*), whose expression appears tightly correlated with *GAMYB* expression [41,46], and corresponds to a class of enzymes which have been associated with PCD and cell lysis. Similarly, inhibition of miR159 in transgenic rice results in up-regulation of pathways associated with PCD, suggesting that *GAMYB* promotes these pathways in the rice grain [61]. Again, what role miR159 plays in regulating GAMYB activity in the seed is unclear. In Arabidopsis germinating seeds, miR159 and *MYB33* are co-transcribed in the aleurone and embryo, however, MYB33 protein is expressed, which suggests miR159 may only be fine-tuning the expression of *MYB33* in this tissue [48]. Nevertheless, *mir159ab* plants produce malformed seeds [14], implying miR159 is required for proper seed development.

### 6.3. The Role of miR159-GAMYB Pathway in Vegetative Tissues

In Arabidopsis, the widespread transcription of *MYB33*/*MYB65*, only to be strongly silenced by miR159 raises the question of what is the purpose of this seemingly futile regulatory pathway. Although miR159 is sometimes associated with leaf development due to the smaller, upwardly curled leaves of the *mir159ab* mutant (Figure 2), this phenotype appears more a consequence of the deleterious impact of *MYB33*/*MYB65* expression rather than the alteration of a developmental program [41]. In general, de-regulated expression of GAMYB in leaves results in strong perturbation of growth. This was shown in *mir159ab*, as well as transgenic Arabidopsis expressing miRNA decoys to inhibit miR159 function, either *MIMIC159* (*MIM159*), Short target tandem *MIMIC159* (*STTM159*), or *SPONGE159* (*SP159*) [58,76], or with *STTM159* rice [61,62], which all result in the similar phenotypic defect of stunted growth. For instance, *STTM159* rice plants are shorter than wild-type rice, with decreased cell numbers, and the most down-regulated genes in *STTM159* rice are associated with cell division. Therefore, the main role of rice miR159 is to suppress *GAMYB* expression to enable cell proliferation [61]. Likewise, the expression of miR159-resistant *GAMYB* transgenes in Arabidopsis [14,16,23,44], lead to the same phenotypic defects. Therefore, it is clear that these *GAMYB* genes encode a class of transcription factors that when expressed inhibit growth, a phenotype contrary to a role in promoting the GA signal for which they were originally identified. Supporting this, GA treatments do not alter the RNA levels of *MYB33*, *MYB65* or miR159 in Arabidopsis rosettes, and the response of *myb33.myb65* plants to GA is not perturbed in vegetative tissues [41]. Therefore, the role the *GAMYB* in transducing the GA signal appears to be tissue dependent, where it is involved in transducing the GA signal in seeds and anthers [70,73,77], but not in vegetative tissues [41]. Supporting this, the Arabidopsis *myb33.myb65* or rice *gamyb* mutants do not appear to have any obvious phenotypic defects at the vegetative stage [43,64].

Contrary to the growth inhibition phenotype of the leaves, the roots of *mir159ab* Arabidopsis are longer than wild-type and have a larger apical meristem zone. Thus, in roots, *GAMYB* expression appears to enhance cell cycle progression, leading to extended roots. However, the root lengths of *myb33.myb65* or *myb33.myb65.myb101* plants were unchanged compared to wild-type, again indicating that these *GAMYB-like* genes are likely fully silenced in roots, again raising the question of what is the role of this pathway in this vegetative tissue [78].

### 6.4. A Role of miR159 in Controlling GA-Mediated Flowering-Time and Growth?

This clear role in inhibiting growth appears at odds with a role often ascribed to *GAMYB* in promoting flowering-time [79,80]. This idea arose from the fact that GA promotes flowering-time, and that the *GAMYB* or *GAMYB-like* genes were thought to be positive regulators of GA throughout the plant [81]. Supporting this idea was the finding that the *LEAFY* gene, a central regulator of flowering, contained a MYB-binding site within its promoter, and this binding site appeared critical in transducing the GA-signal [82]. Subsequently, it was shown that *MYB33* transcription was induced at the shoot apical region upon the induction of flowering, either through GA-application or exposure to long-day conditions, and that the MYB33 protein could bind the *LEAFY* promoter in in vitro gel shift assays [81]. Then, overexpression of miR159 in Arabidopsis [ecotype Landsberg *erecta*, (L*er*)] resulted in down-regulation of *MYB33* expression, which correlated with a decrease in *LEAFY* expression and a delayed flowering-time under short-day conditions [52]. Supporting this is the manipulation of miR159 levels in other plant species that result in altered flowering-times. This includes overexpression of miR159 in rice and wheat which lead to a reduced heading-time [63,66]. Additionally, in the ornamental flowering plant *Gloxinia*, the over-expression of miR159 delayed flowering, whereas the inhibition of miR159 with a *MIM159* transgene accelerated flowering-time [67]. Unlike *MIM159* Arabidopsis or *STTM159* rice [58,59,61,62,76], *MIM159 Gloxinia* did not exhibit any defects in vegetative growth or development [67].

Such evidence argues for a clear and conserved role for miR159 in flowering-time, and that *GAMYB* is likely promoting the GA-signal with regard to flowering. However, overexpression of miR159 in Arabidopsis (ecotype Columbia) did not affect flowering-time [13]. So, although overexpression of miR159 in both ecotypes (L*er* and Columbia) resulted in male sterility due the requirement of GAMYB activity in anthers, there was a differential response with regard to flowering-time. Moreover, a *myb33.myb65* mutant (ecotype Columbia) did not have a delayed flowering-time, and the *mir159ab* mutant (greater GAMYB activity) displayed a late flowering-time under short-day conditions, implying greater GAMYB activity was inhibiting flowering [41]. Given the severe pleiotropic defects of *mir159ab*, it is uncertain whether delayed flowering is a direct result of greater GAMYB activity, or a secondary effect of the severe growth and developmental defects [41].

In addition to delayed flowering, the *mir159ab* Arabidopsis mutant was found to have a strong delay in vegetative phase change (VPC), with the first leaf with abaxial trichomes being *leaf* 16.0 as opposed to *leaf* 7.9 for wild-type, and this was tightly correlated with the increased levels of miR156, one of the key determinants of VPC [60]. In a complex regulatory mechanism, it was found that MYB33 activated transcription of both the *MIR156* gene and its target, *SPL9*, via direct interaction with their promoters [60]. Conversely, Arabidopsis plants overexpressing miR159 (*leaf* 7.3) or the *myb33* mutant (*leaf* 7.1) only had slight increases in VPC compared to wild-type (*leaf* 7.9). This argued that MYB33 protein is expressed to some extent in the Arabidopsis rosette. However, the VPC of a *myb65* mutant (*leaf* 8.1) was unchanged from wild-type, implying *MYB65* did not appear to impact this pathway [60]. Given the subtle changes to vegetative phase change in the *myb33* mutant, the miR159-*GAMYB* pathway was considered a modifier of VPC, where miR159 promotes VPC by preventing MYB33 expression which negatively regulates VPC [60]. It will be interesting to see what role the miR159-*GAMYB* pathway is found to have in this process in other plant species.

Therefore, regarding the miR159-*GAMYB* pathway in growth and flowering, there is strong conflicting evidence. Although the difference in Arabidopsis is possibly due to ecotype variation, a role for GAMYB in either promoting flowering, or alternatively, deleteriously inhibiting growth, will need further experimentation for clarification of how such diametrically opposed outcomes can arise.

### 6.5. Fruit and Reproductive Development

There is growing evidence that the miR159-*GAMYB* pathway plays a role in fruit development. In strawberries, fruit development is GA-regulated, and miR159 is strongly expressed in the fruit’s receptacle tissue and appears to regulate *GAMYB*, as miR159 and *GAMYB* expression is reciprocal [83]. *GAMYB* is a key regulator of strawberry fruit development, as repression of *GAMYB* via RNAi inhibits receptacle ripening and color formation [69]. In tomato, the miR159-*GAMYB* pathway is present in ovules, and overexpression of miR159 resulted in abnormal ovule development, precocious fruit initiation and seedless fruits [35]. Similarly, in grapes, the pathway appears active in the fruits, and under the control of GA [84]. In the monoecious plant cucumber, inhibition of *GAMYB* activity via RNAi altered the ratio of male to female flowers, decreasing the number of nodes with male flowers [68]. Therefore, it appears this pathway is involved in many different functions of the reproductive process in different plant species.

## 7. The Function of the miR159-*MYB* Pathway in Plant Stress

### 7.1. Abiotic Stress

Given the ubiquity and abundance of miR159 throughout the plant kingdom, it is not surprising that numerous studies have implicated miR159 in a wide range of stresses from many different plant species (for review see [85]). In Arabidopsis, miR159 levels increase under salinity [86], and in germinating seeds, miR159 has been found to accumulate in response to the stress hormone ABA as well as to drought [87]. MiR159 also accumulates to higher levels in response to drought in maize, wheat and barley [85]. Such results suggests that increased levels of miR159 may result in greater stress tolerance. However, in some species, miR159 levels decrease in response to drought or salinity [85], and overexpression of miR159 in rice resulted in increased sensitivity to heat-stress [66]. In potato, in which the drought tolerant gene *cap-binding 80* protein has been down-regulated, miR159 levels were decreased and mRNA levels of *GAMYB-like* homologues were higher [88]. Therefore, these studies have found no consistent or unified role for miR159 in plant stress response.

The functional role of the Arabidopsis miR159-*GAMYB* pathway to abiotic stress was investigated by comparing the response a mutant lacking this entire pathway, the *mir159ab*.*myb33*.*myb65* quadruple mutant, to that of wild-type plants [46]. Two-week old plants were exposed to three weeks of treatments with either ABA, high temperature, high light, drought or cold. However, no differential response between the *mir159ab*.*myb33*.*myb65* mutant and wild-type plants were identified. As it was demonstrated that miR159 fully represses *MYB33* and *MYB65* in vegetative tissues of Arabidopsis plants [41], it was rationalized that miR159 levels would need to decrease in Arabidopsis to enable activation of these two *GAMYB-like* genes [46]. However, none of the treatments appeared to repress miR159 to levels in which would allow *MYB33* and *MYB65* expression, and this was supported by assaying the downstream marker gene *CP1*, whose levels appeared completely repressed [46]. Based on this, no clear role for this pathway was identified, and it remains uncertain what role it plays in stress response in Arabidopsis.

### 7.2. Biotic Stress

Similarly, the levels of miR159 respond to many different biotic stresses. Recently it was shown that cotton and Arabidopsis accumulate elevated levels of miR159 in response to the fungus, *Verticillium dahlia* [89]. MiR159 was exported into the fungal hyphae, where it targeted the gene encoding isotrichodermin C-15 hydroxylase (HiC-15), which is critical for hyphal growth. As expression of a miR159-resistant HiC-15 gene in *V. dahlia* resulted in greater virulence, it was concluded that exporting miR159 from the plant was conferring greater pathogenic resistance. Given that the miR159-binding site is highly conserved in HiC-15, it was hypothesized that this has evolved to dampen HiC-15 expression as to avoid rapid death of the host, which then enables establishment of the fungus on the plant [89]. Currently, this is the only clear role for miR159 in pathogen response.

MiR159 also accumulates to higher levels in Arabidopsis root galls that form in response to root knot nematodes (RKN). The *MYB33* gene appears dynamically expressed during gall formation, as a *MYB33: GUS* reporter was expressed during early gall development, but not at later stages. Functional evidence for the involvement of the miR159-*GAMYB* pathway is that an Arabidopsis *mir159abc* triple mutant has greater resistance to root knot nematodes (RKN) [90]. Further investigation will be needed to understand the precise role of the pathway in gall formation and the response pathway to RKN infection.

## 8. Conclusions and Some Unresolved Questions

The miR159-*GAMYB* pathway appears nearly ubiquitous in terrestrial plants, implying it has played an important role in plant’s conquest of the land. Although its role in some tissues now appear to be relatively clear, this is far from the case in others. Below are some of the questions we believe still need to be resolved.

Why are *MYB33* and *MYB65* transcribed in vegetative tissues where failure to fully repress them results in a detrimental effect? What selective advantage does this give the plant?
One hypothesis is that if miR159 is inhibited by a certain trigger, and strong *MYB33*/*65* expression occurs, growth inhibition (or another unknown process) may result in a beneficial outcome (e.g., drought conditions to slow growth). However, currently, no triggers to inhibit miR159 to enable strong *MYB* expression are known.A second hypothesis would be that *MYB33*/*65* are not silenced in all vegetative tissues, but in certain cells they are expressed where they confer a selective advantage. Some evidence suggests *GAMYB* is involved in the transition to flowering, and VPC in Arabidopsis. But currently there is much conflicting data. For instance in Arabidopsis, overexpressing miR159 represses flowering-time, and inhibition of miR159 represses VPC. Other studies have found no role for miR159 in flowering. More work is needed here to clarify these roles, and how conserved they are across species.Why is expression of *GAMYB* in vegetative tissues deleterious and how does it inhibit growth? What down-stream events are these genes triggering? Although some studies have started to address this, more work is needed for a clearer understanding.Is *GAMYB* function related to the way it is regulated, i.e., strongly transcribed, only to then be strongly silenced by miR159? Does miR159 have a role in stress response? Again, many studies have identified changes to miR159 levels in response to a host of different biotic/abiotic stresses, but currently there is no clearly defined role for this miR159 concerning stress tolerance/response.How does the conserved RNA secondary structure associated with the miR159-binding sites of *GAMYB* genes promote their silencing by miR159? Can this structure facilitate a complex regulatory mechanism, enabling strong silencing in some tissues, but poor silencing in others, depending on a dynamic secondary structure configuration? i.e., acting like a riboswitch concerning silencing.What is the role of miR159-mediate regulation on non-*GAMYB* targets? For example, *DUO1* has a conserved miR159-binding site, but the role of miR159 in controlling the expression of this gene remains unclear.What is the role of miR159 in female fertility? Why are Arabidopsis *mir159ab* seeds small and misshapen (likewise rice STTM159 grains are small)? Why does the central cell still divide in some *mir159abc* ovules? How can a seed still form (from *mir159abc* pollen) with a viable embryo when the endosperm divisions stop apparently so early?

## Figures and Tables

**Figure 1 plants-08-00255-f001:**
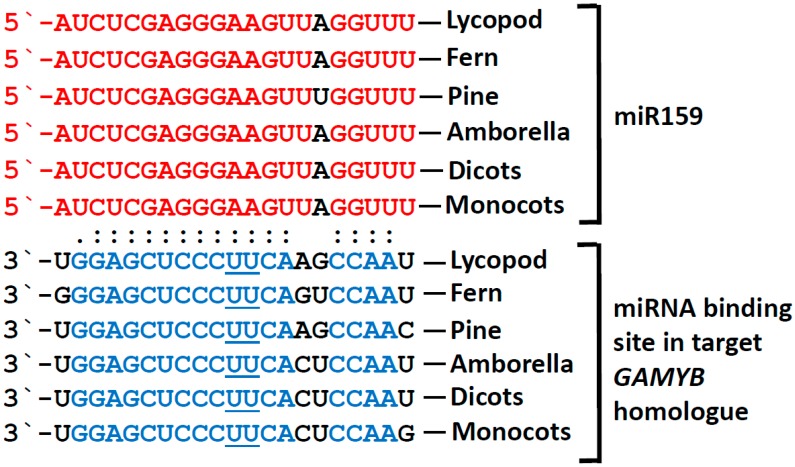
The microR159 (miR159)-*GAMYB* regulatory pathway appears highly conserved in land plants. Similar/identical miR159 isomiRs (shown in red) are found in most plant linages, including Lycopods (*Selaginella uncinata*; [3]) ferns (*Salvinia cucullata*; [3]), pine (*Pinus densata*; [36]), Amborella [37], dicots, and monocots (miRbase; [11]). Highly similar and complementary miR159 binding sites (shown in blue) are found in *GAMYB* homologues from lycopods (*Selaginella moellendorffii*; [20]), ferns (*Salvinia cucullata*; [11]), pine (*Larix kaempferi*; [38]), Amborella and many different monocots and dicots. Variant nucleotide positions are shown in black. However, throughout the plant kingdom, variation is not limited to these positions; for example, see [39].

**Figure 2 plants-08-00255-f002:**
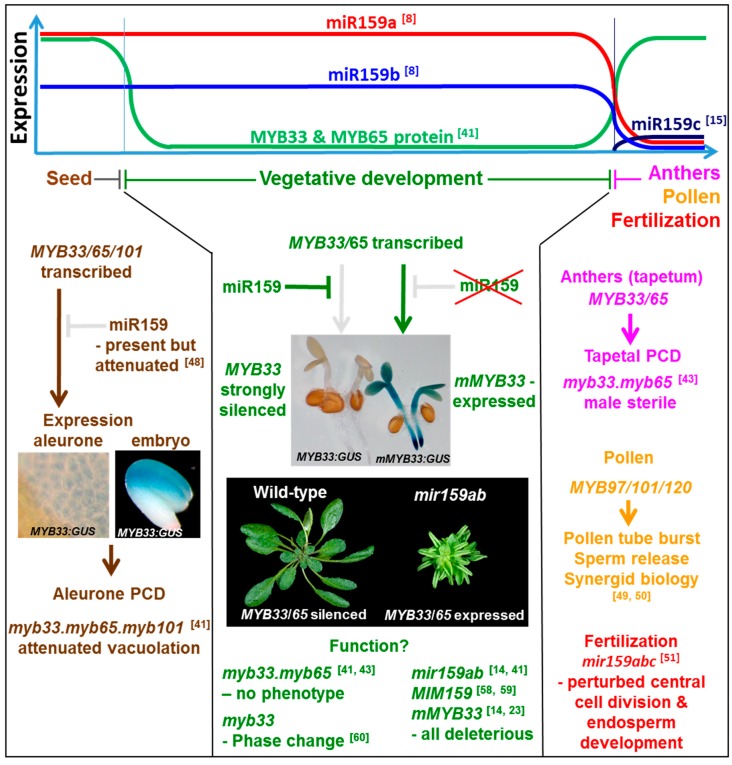
The *miR159-GAMYB-like* pathway in Arabidopsis. miR159a is the predominant family member, being expressed in seed and throughout plant development at a constantly high level, but it is absent in anthers [8,14,46]. miR159b is expressed at a lower level than miR159a [8,15,46], but its expression pattern appears highly similar to miR159a [14]. miR159c, is weakly expressed and appears mainly confined to anthers [15]. In seeds, miR159 efficacy appears attenuated [48], enabling *GAMYB-like* gene expression which promotes PCD of the aleurone [41]. In contrast, throughout vegetative development, miR159 efficacy is strong, and *MYB33*/*65* expression is strongly silenced. Only via inhibition of miR159, or mutation of the miR159 binding site within *MYB33* or *MYB65*, will expression occur, which leads to strong deleterious outcomes, such as stunted growth and curled leaves [14,41]. Although the function of the pathway has been suggested to be involved in flowering-time and phase change, the purpose of this pathway in vegetative development is still unclear. In anthers, miR159 activity is low. Here, *MYB33* and *MYB65* are expressed to promote PCD in the tapetum [43]. *MYB97*/*101*/*120* expression is required for pollen function [49,50]. Finally, miR159 is required for fertilization [51].

**Figure 3 plants-08-00255-f003:**
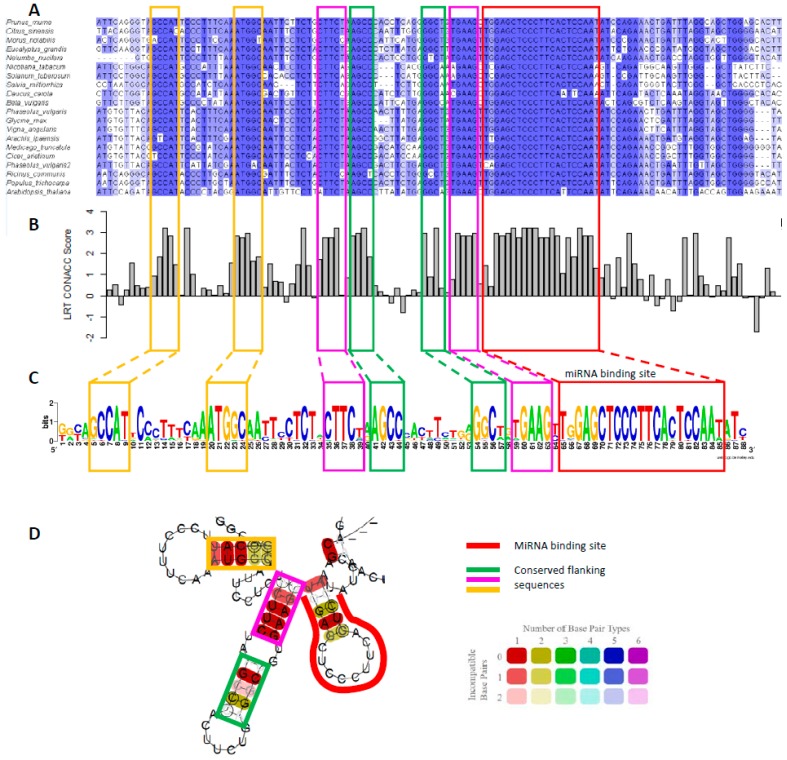
(**A**) Multiple alignment of *MYB33* homologues from different plants species. The binding site is boxed in red, and the conserved flanking sequences in yellow, pink, and green throughout the Figure. (**B**) phyloP score of the multiple sequence alignment of *MYB33* sequences. A positive score denotes evolutionary conservation, whereas, a negative score denotes acceleration [54]. A likelihood ratio test (LRT) was used as the method to detect non-neutral substitution rates. Scores were generated using rPHAST [55]. (**C**) Sequence logo of the binding site and conserved flanking sequences generated using WebLogo [56]. (**D**) RNA secondary structure prediction of the consensus sequence from the multiple alignment in A. generated using RNAalifold [57] at 22 °C and default parameters. Colours represent the number of base pairs types (i.e., AU, UA, CG, GC, UG, GU), and hue the number of non-conserved nucleotides at that position.

**Table 1 plants-08-00255-t001:** miR159 targets in Arabidopsis as determined by different approaches. The top 20 miR159a targets in Arabidopsis as identified by the bioinformatic program psRNATarget with standard search parameters [40]; the number of mismatches is indicated by the score. Confirming this prediction, 5′-RACE analysis can detect miR159-guided cleavage products for at least nine of these genes. In contrast, the more quantitative degradome analysis only identifies three of these genes, with only *MYB33* and *MYB65* being frequently detected in multiple degradome analyses [25,42]. Overexpression of miR159 could detect down-regulation of multiple targets [13,52]. However, genetic analysis using a loss-of-function *mir159ab* mutant identify *MYB33* and *MYB65* as the major important targets [14,41].

	At Number	Score	Name	5’-RACE	Degradome	miR159 OE	*miR159ab*
1	AT4G37770	1.5	*ACS8*	[53]		[13]	
2	AT2G32460	2	*MYB101*	[15,17,53]		[13]	
3	AT3G60460	2	*DUO1*	[15,17,53]			
4	AT2G26950	2	*MYB104*				
5	AT4G26930	2	*MYB97*				
6	AT5G06100	2.5	*MYB33*	[15,23]	[25,42]	[52]	[14,41]
7	AT3G11440	2.5	*MYB65*	[23]	[25,42]		[14,41]
8	AT2G34010	2.5	*MRG1*	[53]	[42]		
9	AT2G21600	2.5	*RER1B*				
10	AT5G55020	2.5	*MYB120*	[15]		[13]	
11	AT4G27330	2.5	*SPL*				
12	AT5G27395	2.5	*Tim44-related*				
13	AT3G61740	3	*SDG14, ATX3*				
14	AT1G29010	3	*MRG-LIKE*				
15	AT4G31240	3	*NRX2*				
16	AT2G26960	3	*MYB81*	[15]			
17	AT2G22810	3	*ACS4*				
18	AT3G08850	3	*RAPTOR1B*				
19	AT5G55930	3.5	*OPT1*	[13]		[13]	[41]
20	AT2G44450	3.5	*beta gluc 15*				

**Table 2 plants-08-00255-t002:** Functional analyses of the miR159-*GAMYB* pathway in plants.

Species	Approach	Phenotype	Ref.
Arabidopsis	T-DNA *mir159ab* mutant	Pleiotropic defects, stunted growth, curled leaves, reduced apical dominance.	[14]
Arabidopsis	T-DNA *mir159c* mutant	none	[15]
Arabidopsis	T-DNA *mir159abc* mutant	Perturbed fertilization	[51]
Arabidopsis	*MIM159* mimic–loss-of-function.	Pleiotropic defects, stunted growth, curled leaves, defective sepals, petals and anthers.	[58,59]
Arabidopsis (Col-0)	miR159a overexpression	Male sterility	[13]
Arabidopsis (Ler)	miR159a overexpression	Male sterility, delayed flowering-time	[52]
Arabidopsis	T-DNA *myb33.myb65* mutant	Male sterile	[43]
Arabidopsis	T-DNA *myb33* mutant	Altered phase change	[60]
Rice	*STTM159* mimic–loss-of-function.	Stunted growth, curled leaves, smaller seeds	[61,62]
Rice	miR159 overexpression	Delayed heading, shorten internode I, malformed flowers, male sterility.	[63]
Rice	*gamyb-1* insertion mutant	Male sterility	[64]
Barley	miR159 overexpression	Male sterility	[65]
Wheat	miR159 overexpression	Delayed heading, male sterility, increased tillering.	[66]
Gloxinia	*MIM159* mimic loss-of-function, miR159 over-expression (OE).	Accelerated flowering (*MIM159*) or delayed flowering (miR159 OE)	[67]
Tomato	miR159 overexpression	Fruit set, parthenocarpy, ovule development, seedless fruits	[35]
Cucumber	RNAi against *GAMYB*	Altered ratio of male to female flowers	[68]
Strawberry	RNAi against *GAMYB*	Inhibition of receptacle ripening	[69]

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
