# Peer review of "Biology and Function of miR159 in Plants"

_plants, 2019, doi:10.3390/plants8080255_

Round 1
Reviewer 1 Report
In this paper „Biology and Function of miR159 in Plants.” Millar and co-workers summarize the latest knowledge about miR159 regulation.
Although being very widespread, expressed abundantly miR159 regulation is still elusive. However, lots of research investigated this question and found very important statements, regulation modes and networks what can show how diverse RNA based regulation can be.
I do believe that a review should be as clear as it could be and unfortunately I found this manuscript very difficult to follow. Maybe a language correction aiming to simplifying descriptions could be a solution.
Strength of the manuscript is the good collection and description of cutting edge researches and I also like the collection of questions at the end what could stimulate following research.
Weakness of the review is that this good collection of information is very difficult to recieve. Figures and Tables should be improved.
As a scientific shortage, I missed a part when role of different miRNA precursors are discussed. MiRNA expression from different loci are spatially and temporally regulated which can add a new layer to their regulation. Sequence complementarity is one aspect of this, but even the same miRNA isoform can be differentially regulate its target because they are differentially expressed.
It would be important to clarify that genome scale degradome analysis are able only to detect cleavage, and cannot correlate miRNAs directly to their targets. This can only be investigated in mutants and overexpression lines.
I think that these statements and descriptions should be corrected before acceptance.
In details I would only suggest modification to the Figures and Tables which would improve the quality and help the readers.
Fig2 is very diverse, it is very difficult to correlate the references with the regulation processes. It is not clear that do the authors have licence for the photos? I would suggest here to place the number of the references into the figure. An alternative solution could be a Table instead of the figure, where according to the tissues effect of the miR159 shortage or overexpression together with the reference could be listed.
Fig 3 is not a figure, but a Table. Instead of Yes the number of the reference could be listed.
I would not include Figure 4 in this review as it doesn’t give any additional information. Instead a paragraph about the differences between plant and animal miRNA regulation could be placed highlighting that in plants the complementarity between the miRNA and their targets are much higher. This is why it is a bit easier to predict real targets for the plant miRNAs than the animal ones. And also to write that bioinformatics analysis doesn’t count with the spatially and temporally different expression of both the miRNA and the target. This could be the reason why predicted targets are difficult to verify.
I miss a table collecting references according to plant species where key finding (miR159 expression, identification of their targets, etc ) are listed together with the references.
As a summary I think that a review about miR159 regulation is a good idea, but before accepting this manuscript it needs scientific and formatting (new Tables and language editing) improvement.
Author Response
Many thanks to the reviewers for the constructive and detailed comments. Below are our point-by-point replies. I have used tracked changes in the manuscript so that the changes can be followed.
Comments
“I do believe that a review should be as clear as it could be and unfortunately I found this manuscript very difficult to follow. Maybe a language correction aiming to simplifying descriptions could be a solution.
Strength of the manuscript is the good collection and description of cutting edge researches and I also like the collection of questions at the end what could stimulate following research.
Weakness of the review is that this good collection of information is very difficult to recieve. Figures and Tables should be improved.”
We have done our best to go through the manuscript and try to simplify the text, shortening sentences and trying to make easier to understand. If the reviewer can give anymore examples that would help. We have revised the figures and presented two new tables.
“As a scientific shortage, I missed a part when role of different miRNA precursors are discussed. MiRNA expression from different loci are spatially and temporally regulated which can add a new layer to their regulation. Sequence complementarity is one aspect of this, but even the same miRNA isoform can be differentially regulate its target because they are differentially expressed.”
This is a good point, and I have added the information on expression of the different miR159 genes in Arabidopsis as determined by promoter:GUS constructs – ‘Examination of their expression domains with promoter:GUS constructs found MIR159a:GUS and MIR159b:GUS had highly similar expression patterns, being broadly expressed throughout the plant, but strongest in shoot and root meristematic regions [14]. By contrast, the expression domain of a MIR159c:GUS reporter gene was much narrower, being restricted mainly to anthers and the shoot apical region [15], suggesting sub-functionalization. To investigate their function, T-DNA loss-of-function mutant alleles were generated for each gene, however, none of the single mir159 mutant plants displayed any phenotypic defects [14-15]. However, consistent with the highly similar expression domains, miR159a and miR159b were demonstrated to be functionally redundant’.
“It would be important to clarify that genome scale degradome analysis are able only to detect cleavage, and cannot correlate miRNAs directly to their targets. This can only be investigated in mutants and overexpression lines.”
We have added a line to address this point. ‘Although this analysis only detects targets regulated by the miRNA-guided cleavage mechanism (and not the translational-repression mechanism), functionally important targets appear to be preferentially detected via degradomes [25, Addo-Quaye et al., 2008].’
I think that these statements and descriptions should be corrected before acceptance.
In details I would only suggest modification to the Figures and Tables which would improve the quality and help the readers.
“Fig2 is very diverse, it is very difficult to correlate the references with the regulation processes. It is not clear that do the authors have licence for the photos? I would suggest here to place the number of the references into the figure. An alternative solution could be a Table instead of the figure, where according to the tissues effect of the miR159 shortage or overexpression together with the reference could be listed.”
We have modified Figure 2 extensively. References have been added in and information about miR159 expression and redundancy. Also a separate table (Table 2) has been construct to include key information.
“Fig 3 is not a figure, but a Table. Instead of Yes the number of the reference could be listed.”
Good point, changed to a table as recommend (Table 1), have updated and expanded table to include the latest psRNATarget search and more extensive degradome analyses. References are now included in the table instead of “yes”.
“I would not include Figure 4 in this review as it doesn’t give any additional information.”
Figure 4 has been deleted.
“Instead a paragraph about the differences between plant and animal miRNA regulation could be placed highlighting that in plants the complementarity between the miRNA and their targets are much higher. This is why it is a bit easier to predict real targets for the plant miRNAs than the animal ones.”
As Figure 4 has been deleted, we do not think such a paragraph would fit smoothly or be of much relevance to this review.
“And also to write that bioinformatics analysis doesn’t count with the spatially and temporally different expression of both the miRNA and the target. This could be the reason why predicted targets are difficult to verify.
Good point, we have added the following sentence ‘Partially explaining this phenomena for miR159, many of the bioinformatically predicted miR159 targets appear to have transcriptional domains that are mutually exclusive to that of miR159, and hence spatially preventing miRNA-target interactions [14-15].’
“I miss a table collecting references according to plant species where key finding (miR159 expression, identification of their targets, etc ) are listed together with the references.”
Good point, table constructed and included (Table 2).
As a summary I think that a review about miR159 regulation is a good idea, but before accepting this manuscript it needs scientific and formatting (new Tables and language editing) improvement.
Reviewer 2 Report
Millar et al. reviewed the biological functions of plant mir159 comprehensively. However, several points still need to be addressed.
1. The contexts related to mir159a, mir159b, and mir159c are mostly referring to the Figure 2 in the manuscript. However, it is difficult to conclude that mir159a and mir159b are functional redundancy in the Figure 2. In addition, it is also not easy to characterize the function of mir159a, mir159b, and mir159c in the Figure 2. Actually, the Figure 2 illustrates the regulation mechanism between MYB33 and mir159. Is it possible to provide another figure that can specify the functions of mir159a, mir159b, and mir159c, respectively?
2. The Figure 3 should be a table, not a figure.
3. I don’t think the Figure 4 is necessary. The number of predicted targets from different species provides limited information. Did the author compare the gene functions of predicted targets among different species?
4. Why select let-7 and lin-4 in the Figure 4? There is no any related information about the relationship among let-7, lin-4, and mir159 in the manuscript. More related background should be provided.
5. It would be better to have a comparison of secondary structure between functional (ex. MYB33 and MYB65) and non-functional (ex. MYB81, MYB97, MYB101, MYB104 et al.) mir159 targets in the Figure 5. If the secondary structures are different between functional and non-functional targets, it could strongly suggest the secondary structure is important for mir159 recognition.
Author Response
Many thanks to the reviewers for the constructive and detailed comments. Below are our point-by-point replies. I have used tracked changes in the manuscript so that the changes can be followed.
Millar et al. reviewed the biological functions of plant mir159 comprehensively. However, several points still need to be addressed.
“1. The contexts related to mir159a, mir159b, and mir159c are mostly referring to the Figure 2 in the manuscript. However, it is difficult to conclude that mir159a and mir159b are functional redundancy in the Figure 2. In addition, it is also not easy to characterize the function of mir159a, mir159b, and mir159c in the Figure 2. Actually, the Figure 2 illustrates the regulation mechanism between MYB33 and mir159. Is it possible to provide another figure that can specify the functions of mir159a, mir159b, and mir159c, respectively?”
We have modified Figure 2 to show where the different miR159 isoforms are expressed to give information on redundancy and specify functions of miR159a, b and c, which is predominantly repressing MYB33 and MYB65 during vegetative development.
“2. The Figure 3 should be a table, not a figure.”
Figure 2 has been changed to Table 1 and has been updated.
“3. I don’t think the Figure 4 is necessary. The number of predicted targets from different species provides limited information. Did the author compare the gene functions of predicted targets among different species?”
Figure 4 has been deleted, as requested by the reviewers.
“4. Why select let-7 and lin-4 in the Figure 4? There is no any related information about the relationship among let-7, lin-4, and mir159 in the manuscript. More related background should be provided.”
As Figure 4 has been deleted as requested by the other reviewers, this information is now not needed.
“5. It would be better to have a comparison of secondary structure between functional (ex. MYB33 and MYB65) and non-functional (ex. MYB81, MYB97, MYB101, MYB104 et al.) mir159 targets in the Figure 5. If the secondary structures are different between functional and non-functional targets, it could strongly suggest the secondary structure is important for mir159 recognition.”
These RNA secondary structures have been published in the reference we have referred to, and therefore we don’t feel they need to be reproduced here. For clarity we have modified the text “Correlating with this difference, is a predicted RNA secondary structure that abuts the miR159 binding site of MYB33 and MYB65, but which was absent in the poorly regulated targets (Figure 5; [41]; also see [41] for RNA secondary structures of the various Arabidopsis GAMYB-like genes).
Reviewer 3 Report
The submitted manuscript reviews the current knowledge on the molecular mechanism and physiological functions of miR159, one of the most conserved miRNA families in land plants. miR159 has been shown to play important roles in a wide variety of physiological phenomenon, and the submitted manuscript would attract a great number of plant physiologists. My biggest concern is that this review is almost entirely focused on miR159-GAMYB pathway because miR159 directly targets and destabilizes GAMYB mRNAs. It has been shown that independent miRNA-Transcription factor (TF) pairs converge on common downstream genes to regulate physiological phenomenon (e.g. miR159-MYB, miR319-TCP and miR167-ARF pairs as described in Rubio-Somoza I and Weigel D (2013) PLoS Genet, e1003374). Therefore, without describing such other pathways, it would be quite difficult to understand the phenotypes that miR159 regulates.
Below are other concerns to be addressed.
Fig.1 implies that miR159 is highly conserved except for one nucleotide, shown in black, in the majority of land plants. However, there are a number of miR159s with some variations in other parts of the nucleotide sequences. (See Pappas Mde C et al. (2015) BMC Genomics, 16:1113 for example). The same holds true for GAMYB mRNA, which are the target of miR159. Those should be clearly stated in the manuscript and also in the figure to avoid misunderstanding and/or confusion. The authors also need to identify the directionality of RNA sequences, that is, which is 5’ and which is 3’.
Fig. 4 is irrelevant because the recognition mechanism is totally different between plant and animal miRNA. It has been widely recognized that a plant miRNA targets mRNA with complementarity of almost entire miRNA region, whereas nucleotides 2 – 8 from the 5’ end of an animal miRNA form the seed, which determines mRNA targets. Also, an animal miRNA targets a much larger number of mRNAs than an plant miRNA does.
Author Response
Many thanks to the reviewers for the constructive and detailed comments. Below are our point-by-point replies. I have used tracked changes in the manuscript so that the changes can be followed.
“The submitted manuscript reviews the current knowledge on the molecular mechanism and physiological functions of miR159, one of the most conserved miRNA families in land plants. miR159 has been shown to play important roles in a wide variety of physiological phenomenon, and the submitted manuscript would attract a great number of plant physiologists. My biggest concern is that this review is almost entirely focused on miR159-GAMYB pathway because miR159 directly targets and destabilizes GAMYB mRNAs. It has been shown that independent miRNA-Transcription factor (TF) pairs converge on common downstream genes to regulate physiological phenomenon (e.g. miR159-MYB, miR319-TCP and miR167-ARF pairs as described in Rubio-Somoza I and Weigel D (2013) PLoS Genet, e1003374). Therefore, without describing such other pathways, it would be quite difficult to understand the phenotypes that miR159 regulates.”
Sorry for the omission. This paper has now been included in the review with the following text “In Arabidopsis, miR159-GAMYB and miR319-TCP pathways in anthers converge, where the GAMYB and TCP proteins directly interact to regulate another miRNA pathway, the miR167-ARF pathway, creating a miR159-miR319-miR167 network that regulates auxin signalling required for developmental progression of anthers”.
Below are other concerns to be addressed.
“Fig.1 implies that miR159 is highly conserved except for one nucleotide, shown in black, in the majority of land plants. However, there are a number of miR159s with some variations in other parts of the nucleotide sequences. (See Pappas Mde C et al. (2015) BMC Genomics, 16:1113 for example).”
The following text is in the review which we believe addresses the reviewers suggestion. ‘Nevertheless, like most miRNA families, considerable variation is found within small RNAs that are defined as being miR159, with most plant species containing multiple family members, that encode identical or highly similar isoforms, or “isomiRs”, that differ by one to several nucleotides. For example, maize has 11 different MIR159 loci, encoding four different miR159 isoforms [12]. For the most part, nucleotide variation occurs at the extremities of the miRNA, at positions considered less important for its specificity [13]. This is the case for the three different miR159 isoforms found in Arabidopsis that vary by 1-2 nucleotides; however, as these three isoforms appear functionally redundant, this variation unlikely impacts which genes they target or their function [14-15]. However, some species have more variant miR159 isoforms (e.g. poplar; grape soybean and maize with 3-5 sequence variations [16]), so whether these miR159 variants have sub-functionalized to regulate different targets remains a possibility.’
Additionally, we have added to the legend of Figure 1; ‘However, it should be noted, throughout the plant kingdom, variation is not limited to these positions; for example see [Pappas Mde et al. (2015)].’
“The same holds true for GAMYB mRNA, which are the target of miR159. Those should be clearly stated in the manuscript and also in the figure to avoid misunderstanding and/or confusion.”
We have added the following; ‘Although there is sequence variation in both the miR159 and its binding site within the GAMYB-like genes….’ and amended the Figure legend.
“The authors also need to identify the directionality of RNA sequences, that is, which is 5’ and which is 3’.”
Directionality has now been indicated on the Figure.
“Fig. 4 is irrelevant because the recognition mechanism is totally different between plant and animal miRNA. It has been widely recognized that a plant miRNA targets mRNA with complementarity of almost entire miRNA region, whereas nucleotides 2 – 8 from the 5’ end of an animal miRNA form the seed, which determines mRNA targets. Also, an animal miRNA targets a much larger number of mRNAs than an plant miRNA does.”
As requested by all reviewers, this Figure has now been deleted.
Round 2
Reviewer 2 Report
The revised manuscript could be accepted.
Author Response
Thanks for your efforts.
Reviewer 3 Report
My biggest concern at the first round of the review, which is no description about the interconnection with other miRNA/TF pathways than miR159-MYB, has not been improved. A few sentences have been added since the initial version of the manuscript, but it is still far from sufficient level for molecular- and genetic-level understanding of the role of miR159.
Author Response
Thanks for the comments and effort.
Reviewer comment.
"My biggest concern at the first round of the review, which is no description about the interconnection with other miRNA/TF pathways than miR159-MYB, has not been improved. A few sentences have been added since the initial version of the manuscript, but it is still far from sufficient level for molecular- and genetic-level understanding of the role of miR159."
I have expanded the description of the interaction of miR159 with miR319 and miR167 into a full paragraph and added some analysis.
‘In Arabidopsis, the role of the miR159-GAMYB pathway and its interaction with the miR319-TCP pathway has been investigated in flower maturation using miRNA loss-of-function MIM159 and MIM319 transgenic plants, both of which display multiple pleiotropic defects [54]. In sepals, petals and anthers of these plants, it was found that the GAMYB and TCP proteins are expressed and directly interact to regulate another miRNA, miR167, which creates a miR159-miR319-miR167 network. It is proposed that the function of miR159/miR319 is to dampen MYB/TCP expression, resulting in low miR167, and hence strong ARF6/8 expression, which in turn regulates many genes required for floral development including that of auxin signalling [55]. However, in wild-type plants, it appears MYB33/65 expression in flowers is restricted to anthers, and the myb33.myb65 mutant only displays anther defects [39]. So in wild-type, it appears the role of miR159 in flowers is to strongly repress the MYB genes in sepals and petals to prevent strong expression of miR167 to ultimately enable strong ARF6/8 expression.’
Also the review describes the interaction between miR159 and miR156.
‘In a complex regulatory mechanism, it was found that MYB33 activated transcription of both the MIR156 gene and its target, SPL9, via direct interaction with their promoters [72].’
Both papers describe complex pathways via multiple experiments, and I have endeavored to get the big picture messages out of these papers. I don’t I think covering the complex details from these papers will add anything to the review which is trying to summarize dozens of different papers to make people aware of the various studies that have been performed.
Round 3
Reviewer 3 Report
I think that the current version of the manuscript sufficiently describes molecular functions and physiological roles of miR159 in plants.